# ‘It’s All Downhill from Here’: A Scoping Review of Sports-Related Concussion (SRC) Protocols in Downhill Mountain Biking (DHI), with Recommendations for SRC Policy in Professional DMB

**DOI:** 10.3390/ijerph191912281

**Published:** 2022-09-27

**Authors:** Michael McLarnon, Stephen H. Boyce, Neil Fisher, Neil Heron

**Affiliations:** 1Belfast Health and Social Care Trust, Belfast BT9 7AB, UK; 2Emergency Department, Glasgow Royal Infirmary, Glasgow G4 0SF, UK; 3Scottish Institute of Sport, Stirling FK9 5PH, UK; 4UK Athletics, Birmingham B42 2BE, UK; 5British Cycling, Manchester M11 4DQ, UK; 6Centre for Public Health Research, Queen’s University, Belfast BT7 1NN, UK; 7Department of General Practice, Keele University, Newcastle ST5 5BG, UK

**Keywords:** sports-related concussion, concussion, downhill mountain bike (DMB), head trauma assessment, consensus meeting

## Abstract

Introduction: Downhill mountain biking (DHI) is a form of cycling and does not currently have a specific sports-related concussion (SRC) assessment. Objective: To review the extent, range and nature of research investigating SRC in DMB, provide a summary of key literature findings relating to its identification and management, and then develop a SRC protocol specific to DMB. Design: Scoping review as per recognised methods. Setting: Literature-based. The following databases were searched: MEDLINE, EMBASE, Scopus and Web of Science, with no restrictions on date. Results were limited to the English language. Participants: Six articles were included in the review from 64 identified articles. The article had to specifically include an analysis of adult downhill riders for inclusion. Outcome measures: Study type, study group (amateur/professional), concussion incidence, concussion assessment and recommendations. Main Results: Concussion incidence was identified as between 5–23%. No study outlined a trackside assessment of cyclists or a protocol for return to play where SRC was identified. Several authors identified that riders often continued to participate despite the presence of a concussion. No sport-specific SRC assessment was determined for DHI, and a SRC assessment was therefore developed. Conclusions: This review illustrates the lack of studies and formal protocol in SRC assessment for DHI. In light of this, we propose a three-stage framework specific to the sport to best identify a concussion and act where appropriate while minimising disruption to competition. This framework involves assessing the cyclist on the ‘sideline’, a second assessment post-event in the medical room and a third assessment the following day. A SRC consensus meeting specific for DHI is suggested with an identified need for updated guidance from UCI, requiring possible rule changes for the sport.

## 1. Introduction

Sports-related concussion (SRC) is a form of traumatic brain injury defined as a complex pathophysiological process affecting the brain [1]. It is induced by biomechanical forces and has several common characteristic signs and symptoms. Despite this, limited tools aid its diagnosis or management [1]. A concussion is a growing concern from a public health perspective and accounts for many injuries within both amateur and professional cycling [2,3] Recent tools have been developed to identify and manage SRC in road [4,5,6,7] and track cycling [8] However, downhill racing is an arguably higher risk sport with regards to traumatic injury and indeed head injury [9,10] but does not have any formal concussion policy. 

Downhill mountain biking is a form of cycling, specifically a genre of mountain biking [11] Races are in a time-trial format, with cyclists starting at intervals or in a mass start and the aim is to complete a downhill course in the fastest time possible whilst remaining between two strips of tape demarcating the route [12]. Riders practice and then do qualifying runs, which then places them for the final one or two runs. Each rider must complete at least two practice runs before starting their qualifying runs. 

The exact incidence of concussion in sport is poorly defined in general as many cases are thought to go unreported [13]. It is likely, at least partially for this reason, that the incidence is increasing in cycling alongside an increased awareness and assessment of concussion, which has one of the highest incidence of SRC of any sport [14,15]. Downhill mountain biking has previously been identified as an extreme sport with a high risk of serious injury [10,16,17,18,19]; one prospective study identifies a risk of concussion in participants of 5% [16], and another large study of mountain biking hospital admissions to a major trauma centre reports head injuries at 12%, with concussion the commonest suspected diagnosis [19]. Specifically, by nature of course design, DHI exposes cyclists to a high number and magnitude of accelerations; this can affect rider executive function in and of itself, but also creates an environment with high potential for SRC injuries [9,10]

Identification of SRC is paramount for an athlete’s physical and mental health to avoid further injury (including the risk of second impact syndrome) [20] and to then determine the athlete’s safe return to play and sport. Indeed, a failure to identify SRC, or a premature return to play following a concussion, can expose a cyclist to an increased risk of SRC reinjury, a more severe form of SRC injury, prolonged physiological symptoms and/or other musculoskeletal injury [21,22,23,24]. Failure to identify or undertake a graduated return from SRC to sport can also lead to increased injury risk for other competitors. In the long term, evidence indicates that SRC, and more specifically repeated SRC, can lead to depression and cognitive deficit in later life, in a recognised condition known as Chronic Traumatic Encephalopathy [23,25]. 

Furthermore, there is further evidence suggesting that even where SRC is identified, some cyclists continue to compete, placing themselves at higher risk of subsequent injury [16]. Downhill mountain biking is a sport at the more extreme end of the competitive cycling spectrum in terms of the potential risk of injury and, indeed concussion. It is therefore important to investigate current guidance for the management of SRC within this sport and, if absent, make sport-specific suggestions towards a future protocol for identification and graduated return to play.

## 2. Aims

The aims of this scoping review are to:Review the extent, range and nature of research that has investigated SRC in downhill mountain biking;Provide a narrative summary of the key literature findings in relation to methods for identifying and managing SRC in downhill mountain biking (including a return to play recommendations); and,Where absent, develop protocols to identify and treat SRC in downhill mountain biking.

## 3. Methods

A scoping review was considered the most appropriate methodological approach to address the research question. A scoping review is defined as an exploratory project that systematically maps the literature available on a topic, identifying key concepts, theories, sources of evidence and gaps in the literature [26,27]. Compared to systemic reviews, it allows us to investigate a less specific research question and acts to inform future research [28]. This scoping review was based on the five-step methodological framework outlined by Arksey and O’ Malley and informed by Levac et al. [29,30]. The Preferred Reporting Items for Systemic Review and Meta-Analysis (PRISMA) was followed, and the PRISMA extension for scoping reviews (PRISMA- ScR) checklist was completed [26].


*Stage 1: Identify the Research Question(s)*


The following research question will be addressed in this report:

1. Does any guidance exist for the identification or management of sports-related concussions in downhill mountain biking?


*Stage 2: Identifying relevant studies*


Eligibility criteria

The review authors decided on the following inclusion and exclusion criteria. 

Inclusion criteria:Specific population group: cyclists, recreational and professional/elite;Participants aged 18 years or above and all sexes;Participants from downhill mountain biking;Research that addresses any aspect of concussion-related to the sport;Sources of information could include primary research studies, reviews (including but not limited to scoping reviews, systematic reviews, and meta-analysis), case reports, conference papers;Articles written in English.

Exclusion criteria:
Studies not published in English;Inability to access the full-text research article;Opinion pieces/opinions, commentaries, consensus papers, magazine, and newspaper articles;Articles not directly related to downhill mountain biking.

### Search Strategy and Databases

To identify relevant studies, a literature search was conducted, with the most recent search on 31 May 2022, with no date limit, during which the following electronic databases were searched: MEDLINE, EMBASE, Web of Science and Scopus. Detailed search strategies for each database were decided a priori. The final search strategy is detailed in Appendix A. 

Further hand-searching was performed on the bibliographies of the eligible studies. The search results were imported to Endnote Reference Manager (Version X9.3.3), where duplicates were removed.


*Stage 3: Study selection*


The titles and abstracts of the studies obtained by the search strategy were independently reviewed against the eligibility criteria by the two authors (NH and MM). Based on their titles and abstracts, the studies were grouped as ‘include’, ‘exclude’, or ‘uncertain’. Articles that did not meet the eligibility criteria were discarded, and their reasons for exclusion were clearly documented. Any disagreements on study selection were resolved through discussion between the two reviewers, with a third reviewer available if required. The full texts of all the included and ‘uncertain’ publications were retrieved, and their contents were screened further against the eligibility criteria.


*Stage 4: Charting the data*


Data elements were extracted by one reviewer (MM), and a second reviewer (NH) checked the accuracy of all of MM’s data extraction. Any issues with data extraction were resolved at a meeting between the two reviewers. Data extraction categories included:Author;Year of publication;Journal/source of publication;Aims/purpose of the study;Study populations, including the performance level of the athletes, included in the study; Study design; Intervention type and duration (if applicable);Outcome measures and details of these (if applicable);Important results related to the scoping review research question. 


*Stage 5: Collating, summarising, and reporting the results*


Through collating, summarising, and reporting the results, we provided a concise summary of existing research findings relating to SRC in downhill mountain biking and identified research gaps. As per protocol, the results were presented in a narrative synthesis to summarise the main findings of the review. We grouped the studies by outcome utilised to monitor for SRC and provided a descriptive summary of the findings. There was no assessment of methodological limitations, risk of bias, or any meta-analysis of the data in this scoping review. The research team then applied the results to downhill mountain biking to develop a protocol to assess for SRC and subsequently manage it within this specific at-risk population.

## 4. Results

### 4.1. Identification of Studies

An overview of the study selection process is provided in Figure 1. The initial literature search identified 64 records, with 57 remaining after duplicates were removed. Forty-seven were removed based on their title. The article had to include an analysis (or sub-analysis) of downhill mountain bike riders for inclusion. Three further articles were removed on abstract screening, leaving eight articles to be reviewed in full. Four full-text articles were excluded for the following reasons: no full-text article was available for one, and another was not in English; two ultimately were review articles with no guidance on concussion. An additional two records were identified through hand searching the reference lists of the eligible studies. Six records were identified as relevant to the aims and research question and included in the analysis.

### 4.2. Study Characteristics

The publication types of the included studies, sport, level of athletic participation studied, incidence of concussion, tools used for assessment of concussion and recommendations made are included in Table 1.

## 5. Discussion

This is the first scoping review documenting the incidence of concussion in DHI and identifying the current lack of sport-specific protocols for concussion assessment and return to play within the sport of DHI. We identified six eligible studies for the review, which revealed a high incidence of concussion of between 5–23% in downhill mountain biking. One article (Clark et al., 2021) [34] used the SCAT-5 tool to identify concussions, but no study outlined a ‘trackside’ or ‘pitch side’ assessment of downhill mountain bikers or a protocol for return to play when concussion was identified. Willick et al., 2021 [35] alluded to a minimum four-week exclusion period in cyclists with a confirmed concussion, whilst Clark et al., 2021 [34] concluded that there is a need for mandatory sideline concussion protocols and increased cyclist education surrounding concussion.

### 5.1. Concussion Incidence

A lower incidence in earlier studies could be explained by the well-documented lack of identification and awareness surrounding concussion and its assessment at the time, with an increased incidence across all sports (including professional cycling) recognised in more recent years [36,37,38]. Concussion incidence in the included studies is high for DHI, reported being between 5–23% of all injuries. Worryingly, Clark et al., 2021 [34] identified a high propensity for athletes to continue cycling despite experiencing concussion symptoms, and athletes should be aware of the need to be removed immediately from the competition when a concussion is suspected, in keeping with other sports, such as road cycling [38], and the message, ‘if in doubt, sit them out’ [39]. This message encourages sports teams, athletes and coaches to err on the side of caution, removing an athlete from play in any event where an SRC has even potentially occurred. The highest incidence was observed in the prospective study by Willick et al., 2021 [35], where athletes were assessed at the point of injury, although no formal sideline assessment was documented.

### 5.2. Current Guidance for Concussion Assessment

The UCI currently recommends the use of the SCAT-5 tool (mentioned by Clark et al., 2021) for concussion assessment [34]. However, this has previously been identified as impractical and insufficient in road cycling, with this tool needing to be amended for the demands of the specific sport. The SCAT-5 questionnaire specifically states that it cannot be performed adequately in less than ten minutes [40] which can render it non-viable during competition, particularly within a sport such as DHI (where races typically last 2–5 min total) [11]. A recently proposed alternative for concussion assessment in road cycling, the RIDE assessment, has been proposed [6]. However, there is currently no sport-specific guidance for DHI, so we propose modifications of the RIDE assessment [6] to be used in downhill mountain biking.

### 5.3. A Framework for Managing SRC in Downhill Mountain Biking: RIDE-DHI

The nature and rules of the sport [12] must be considered when formulating a protocol for assessing SRC, and we propose the modified RIDE-DHI protocol specifically for professional downhill mountain biking.

Whenever a potential head injury occurs with risk of SRC, the cyclist should immediately be removed from the course and assessed on the sideline by key team staff or independent race doctors. The potential SRC event can be observed either directly or via video review (where available), and key personnel should be spread across the racetrack. The cyclist should then be assessed using the RIDE-1 DHI assessment, which will include:(a)An assessment for the presence or absence of 12 Immediate and Permanent Removal features that, if present, warrant immediate and permanent withdrawal from the competition;(b)In the absence of any of the 12 Immediate and Permanent Removal features, a standardised roadside screening assessment, including symptom checklist, medical evaluation, balance assessment and cognitive tests performed by the race doctor and/or team doctor;(c)Clinical evaluation by the race doctor and/or team doctor.

If any of the 12 Immediate and Permanent Removal features are identified at this stage, the cyclist is immediately and permanently removed from the competition:
1Convulsion;2Tonic Posturing;3Suspected loss of consciousness;4Confirmed loss of consciousness;5Cleary dazed;6Ataxia;7Oculomotor signs;8Rider not oriented in time, person or place (TTP) or fails to complete orientation questions;9Definite confusion;10Definite behaviour change;11Identification of any sign or symptom of concussion;12High risk features.


Those who have sustained a potential SRC event should undergo a standardised assessment at the side of the track using the RIDE-DHI 1 protocol by appropriately trained medical staff (Figure 2), with race marshals ‘holding’ the rider in place until the medical staff are on scene. Suggested Modified Maddock’s questions to use for downhill mountain biking are:
○What is the name of this race or track?○How far are you from the finish line?○What is the name of your racing team? An alternative if not part of a race team: where is the race taking place (city and country)?○What position are you seeded/what is your order in this race?○What is your coach’s name?
Figure 2Ride-DHI 1.
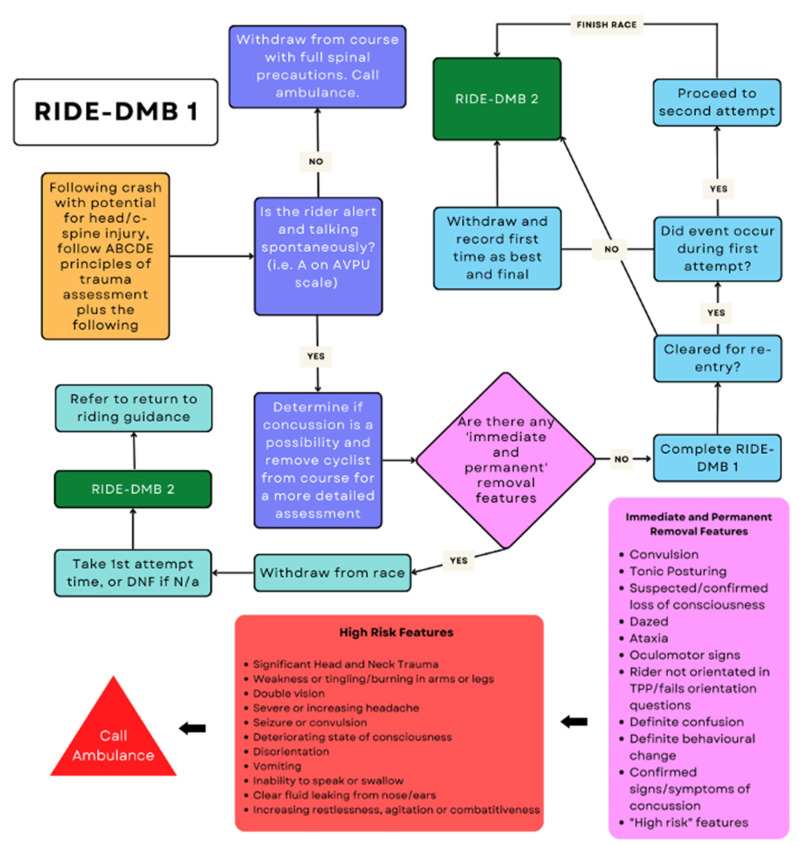


If a rider is cleared for competition following this assessment, they can make their second attempt, or if the potential event occurred during the qualification round, they can continue in the competition. If the athlete is diagnosed with a concussion on their second attempt, then their first attempt should be their final time or if on their first attempt, then use their fastest qualifying time. 

For all riders sustaining a potential SRC during competition but who have ‘passed’ HIA-DHI 1 (Figure 2), a further two assessments should occur post-event. The first of these subsequent assessments (RIDE-DHI 2) (Figure 3) is performed immediately post-race, in the medical room, away from noise and distractions, with the intention of assessing clinical progress and signs of developing SRC. This is a full assessment by a qualified doctor per the UCI recommended SCAT-5 questionnaire, in addition to completing neuro-cognitive tools such as Digit Symbol Substitution Testing (DSST) and a full neurological exam. All riders should have baseline screening data obtained pre-season to both inform and guide decision-making. This baseline screening should be done by the team medical staff, if available, or by the event medical team at the 1st race of the season and stored in an appropriate, confidential but accessible location for medical staff to access. The UCI should help with the safe storage of this medical information. 

The final assessment (RIDE-DHI 3) (Figure 3) should be performed the next day in order to identify a late presentation of SRC. This should be a full clinical assessment with the same components as RIDE-DHI 2: the formatted SCAT5, neuro-cognitive tools such as DSST, and a full neurological exam.

Only following the successful completion of all three RIDE-DHI components should riders be cleared to commence further training and competition. A diagnosis of SRC can be made at any stage following identification of any of the 12 Immediate and Permanent Removal features following a head impact event. In the absence of any of the 12 Immediate and Permanent Removal features, the diagnosis cannot be excluded until both RIDE-DHI 2 and RIDE-DHI 3 assessments are completed and deemed normal. Where a diagnosis of SRC is made, return to riding guidelines should be followed as per SCAT-5. 

Outlined below are suggested Return to Riding guidelines, adapted from the Fifth International Conference on Concussion in Sport (2017) (Table 2) [21]. These should be adapted/modified in a discussion between the team physician, coach and athlete. Where an athlete fails to complete a stage or symptoms return within that stage, they should not progress onwards. Cyclists should remain in each stage for a minimum of 1 day before progression. Delayed completion of the protocol (>4 weeks) will require further investigation and additional rehabilitation, co-ordinated by the team doctor where appropriately trained, or an expert physician in concussion.

Note the importance of rest prior to the commencement of Stage 1. The risk of re-injury is greatest during the first week following concussion, and therefore rest and low-level activity attenuates this risk [41]. If the cyclist engages in intense bouts of intense physical activity during this period, this may temporarily exacerbate their symptoms [41,42]. Until they are symptom-free, the cyclist should not consume alcohol or drive a motorised vehicle and should be kept under strict supervision in the first 24 h for emergent symptoms. When providing an estimated timeline, approximately 80% of athletes will progress to stage 6 and be fully recovered within three weeks.

Implementing the RIDE-DHI protocol would require rule changes from the UCI regarding their downhill mountain biking policy. We suggest this should be in tandem with an educational programme delivered for riders and their teams around concussion and how it will be assessed and managed for them.

### 5.4. Future Direction

With emergent evidence highlighting the magnitude of potential head impacts and their potential impairment of cognitive function (even without falling from the bike) [9,10], future technology could incorporate accelerometers on helmets that sense when potential SRC events occur.

These could be applied in two ways. Firstly, pre-testing courses for DHI and limiting the number of rotational/translational accelerations that occur in any given race (or the total value in grams of significant accelerations (>10 g) [10] to minimise acceptable injury risk may be an option; this would require further research into effects of potential SRC injury from cumulative low energy accelerations (with current evidence indicating reduced executive functioning by this mechanism in DHI) and quantifying cut-offs [9].

Alternatively, suppose cut-off/high-risk acceleration values were identified during races. In that case, this may provide evidence for removing a cyclist where ambiguity exists or encourage further post-race assessment.

## 6. Conclusions

This is the first review of its kind to investigate the available evidence of SRC assessment in downhill mountain biking, illustrating the lack of studies and formal protocols in this area. In light of this, we propose a three-stage framework specific to the sport of downhill mountain biking in order to best identify a concussion and act where appropriate whilst minimising disruption to competition. This framework involves assessing the cyclist on the sideline before a potential second attempt (or taking their first attempt where the injury occurs on the second) (RIDE-DHI 1), a second, more thorough assessment post-event (RIDE-DHI 2), and a third assessment the following day (RIDE-DHI 3). We also recommend an exclusion period from competitive sport following SRC, in keeping with other sport SRC guidelines, with a graded return to cycling. A SRC consensus meeting specific for downhill mountain biking is suggested, with an identified need for updated guidance from the UCI, requiring possible rule changes.

## Figures and Tables

**Figure 1 ijerph-19-12281-f001:**
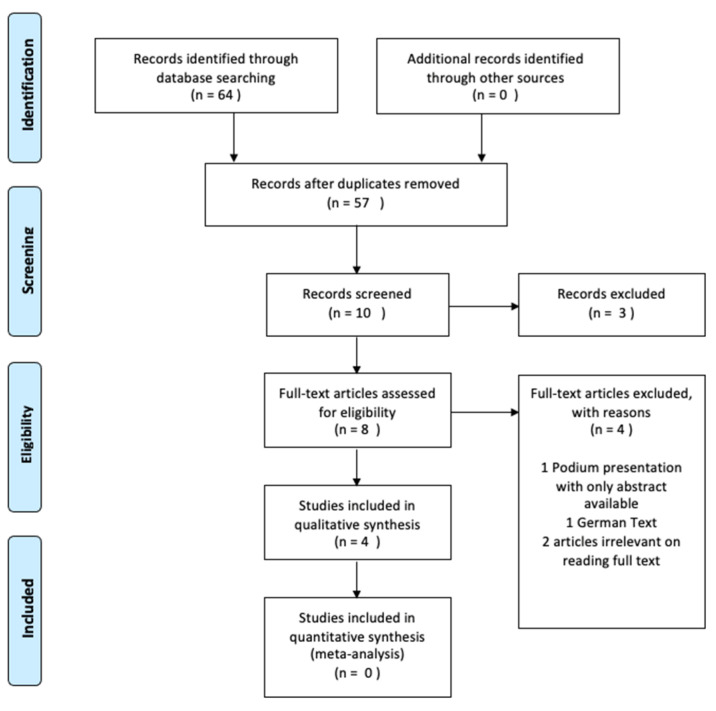
Prisma Flow Diagram.

**Figure 3 ijerph-19-12281-f003:**
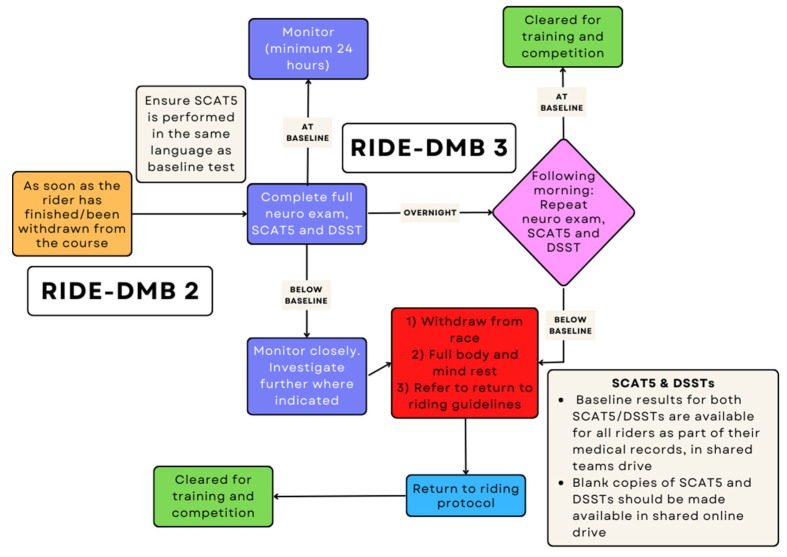
RIDE-DMB 2 and 3.

**Table 1 ijerph-19-12281-t001:** Characteristics of Included Studies.

Author, Year	Study Title	Type of Study	Sport	Amateur/ Professional	Concussion Incidence	Concussion Assessment	Continued to Participate?	Recommendations	Notes
Kronisch, 1994 [31]	Traumatic Injuries in Off-Road Cycling	Retrospective, cross-sectional survey	Cross-country, downhill and dual slalom mountain biking	Amateur	5.6% (3/54)	-	-	-	-
Kronisch, 1996 [32]	Acute Injuries in Off-Road Bicycle Racinig	Descriptive, prospective	Off-road/downhill mountain biking	Professional	25% (4/16) of those injured overall (3624)	-	-	-	5/834 specific to downhill injured overall. Most serious injuries occurred in DHI.
Kronisch et al., 1996 [33]	Acute Injuries in Cross-Country and Downhill Off-Road Bicycle Racing	Retrospective Analysis	Downhill mountain biking	Professional	9% (1/11)	No	-	None	-
Becker et al., 2013 [16]	A Prospective Study of Downhill Mountain Biking Injuries	Prospective survey	Downhill mountain biking	Amateur and Professional	5% (23/494) 249 cyclists, 494 injuries	-	2 cyclists with concussion	Improved protective equipment and safer downhill trails	Higher injury rates among professionals
Clark et al., 2021 [34]	Do Mountain Bikers Know When They Have Had a Concussion and, Do They Know to Stop Riding?	Retrospective Survey	Downhill mountain biking	Professional	6.9% (15/219) diagnosed 12.8% (28/219) concussion symptoms OR 1.24 for downhill vs. cross-country	SCAT 5 via anonymous survey post-competition	67.5% Yes	Event organizers should implement mandatory sideline concussion protocols & encourage mountain bike advocacy groups to develop concussion sensitization programs that will educate riders	1/3 did not recognise concussion 5× risk of concussion in sponsored athletes
Willick et al., 2021 [35]	The National Interscholastic Cycling Association Mountain Biking Injury Surveillance System: 40,000 Student-Athlete-Years of Data	Prospective study	Downhill mountain biking	Amateur (Student Athlete)	23.6%	Not documented	Not documented- 71.3% of all injuries were unable to continue (not only head injury)	None- alluded to ≥4 weeks exclusion for confirmed concussion	55.5% of all injuries occurred while riding downhill Concussion commonest injury

Six texts were ultimately included, all discussing SRC in downhill mountain biking. Three included studies reported information regarding the incidence of concussion in downhill cycling, including one (Clark et al., 2021) [34] noting that the diagnosis is often missed and is more common in sponsored athletes. None provided any information on a ‘sideline’ assessment for concussion or recommendations for return to play. Three were professional competitors, two were amateur cyclists, and one involved both.

**Table 2 ijerph-19-12281-t002:** Return to Riding guidelines, adapted from the Fifth International Conference on Concussion in Sport.

Stage	Aim	Activity	Goal
Initial Rest Period. Should Be Symptom-Free and Have Returned to School/Work/University Activities before Progressing to Stage 1.			
1	Symptom-limited activity	Daily non-symptom-provoking activities	Gradual reintroduction of work/school activities
2	Light aerobic exercise	<15 min of duration. Walking or stationary cycling at a slow to medium pace. No resistance training.	Increase heart rate. Heart rate to be less than 70% of maximum.
3	Sport-specific exercise	Less than 45 min in duration. Cycling drills on smooth terrain. Sprints on stationary bike.	Add movement. Heart rate to be less than 80% of maximum.
4	Non-contact training drills	Less than 60 min in duration. Harder training drills, e.g., introduce downhill terrain at submaximal effort or obstacle courses/sharp turns on smooth terrain. Begin progressive resistance training.	Exercise, co-ordination and increased thinking. Heart rate to be less than 90% of maximum.
5	Full contact practice	Following medical clearance, participate in normal training activity	Restore confidence and assess functional skills by coaching staff.
6	Return to sport	Normal mountain bike training and competition.	

## Data Availability

The data presented in this study are available on request from the corresponding author.

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
