# Peer review of "‘It’s All Downhill from Here’: A Scoping Review of Sports-Related Concussion (SRC) Protocols in Downhill Mountain Biking (DHI), with Recommendations for SRC Policy in Professional DMB"

_ijerph, 2022, doi:10.3390/ijerph191912281_

Round 1
Reviewer 1 Report
General comments
This scoping review focuses on the development of a suitable SRC protocol for use in Downhill Mountain Biking (DHI). Overall, the review is well written, though reference to several pertinent studies on head injuries and DHI (see comments below) are missing. With respect to the proposed SRC assessment protocol, I think the authors suggestion are logical and implementable, though this may required some rule changes by the UCI.
Specific comments
Abstract:
L18 – The UCI uses the abbreviation ‘DHI’ for Downhill Mountain Biking. Please use this abbreviation throughout, to be consistent.
L22 – Please include the dates or reference numbers for Arksey and O’Malley and Levac et al.
Introduction:
The introduction is well written and presents the scope of the problem and potential risks of SRC in cycling well. However, the authors make no reference to published work that has directly assessed the magnitude of head accelerations or the role these may play in SRC and cognitive decline. Therefore, it is recommended the authors should make reference to Hurst, HT., Hancock, S., Hardwicke, J. and Anderson, E. (2020) Does participation in Downhill mountain biking affect measures of executive function? Journal of Science and Cycling, 9(3), 83-93 and Hurst, HT., Atkins, S. and Dickinson, DB. (2018) The Magnitude of Translational and Rotational Head Accelerations Experienced by Riders During Downhill Mountain Biking. Journal of Science and Medicine in Sports, 21(12), 1256-1261. This would add support to your statement of DHI being at the extreme end of the cycling spectrum and the increased risk of SRC.
Methods:
The methods state a 6-step process was followed. However, only 5 stages are presented in the methods. Please could the authors clarify which is correct.
Discussion:
L216 – It may be useful for the authors to clarify the typical duration of a DHI race, and hence the lack of utility for the SCAT-5 in such races.
Please be consistent with the use of either RIDE-1 DMB or RIDE-DMB 1.
L264 – What/where was your last race? This may not be an appropriate question to ask, as the race official doing the assessment may not know what/where this was for that rider.
L266 – Suggest removing the question ‘What is in your jersey (back) pockets?’ as DHI jerseys do not have pockets at all.
Based on current research by Hurst et al (2018; 2020) on the magnitude of head impacts in DHI and the potential to reduce cognitive function, could the authors provide some brief commentary on the potential role of technology in monitoring head impacts during DHI and its possible utility in decision making as to when to stop a rider for assessment?
L275 – Should ‘HIA-DMB 1’ read ‘RIDE-DNB 1’?
Conclusions:
Please refer to RIDE-DMB to be consistent.
Author Response
Reviewer 1
General comments
This scoping review focuses on the development of a suitable SRC protocol for use in Downhill Mountain Biking (DHI). Overall, the review is well written, though reference to several pertinent studies on head injuries and DHI (see comments below) are missing. With respect to the proposed SRC assessment protocol, I think the authors suggestion are logical and implementable, though this may required some rule changes by the UCI.
Specific comments
Abstract:
L18 – The UCI uses the abbreviation ‘DHI’ for Downhill Mountain Biking. Please use this abbreviation throughout, to be consistent.
This has been corrected throughout the text to now read “DHI”.
L22 – Please include the dates or reference numbers for Arksey and O’Malley and Levac et al.
This has been corrected:
Arksey and O’Malley., (2005)
Levac et al., (2010)
We added the dates as opposed to references to avoid references in the text of the abstract.
Introduction:
The introduction is well written and presents the scope of the problem and potential risks of SRC in cycling well. However, the authors make no reference to published work that has directly assessed the magnitude of head accelerations or the role these may play in SRC and cognitive decline. Therefore, it is recommended the authors should make reference to Hurst, HT., Hancock, S., Hardwicke, J. and Anderson, E. (2020) Does participation in Downhill mountain biking affect measures of executive function? Journal of Science and Cycling, 9(3), 83-93 and Hurst, HT., Atkins, S. and Dickinson, DB. (2018) The Magnitude of Translational and Rotational Head Accelerations Experienced by Riders During Downhill Mountain Biking. Journal of Science and Medicine in Sports, 21(12), 1256-1261. This would add support to your statement of DHI being at the extreme end of the cycling spectrum and the increased risk of SRC.
Thank you for referencing these papers- we believe they have improved the quality of our introduction.
Reference 9 is now: Hurst H, Hancock S, Hardwicke J, Anderson E. Does participation in Downhill mountain biking affect measures of executive function? Journal Of Science & Cycling. 2021;9:83-93.
Reference 10 is now: Hurst HT, Atkins S, Dickinson BD. The magnitude of translational and rotational head accelerations experienced by riders during downhill mountain biking. J Sci Med Sport. 2018;21(12):1256-61.
We have added the references at the following points:
“However, downhill racing is an arguably higher risk sport with regards to traumatic injury, and indeed head injury,(9, 10) but does not have any formal concussion policy.”
“Downhill mountain biking has previously been identified as an extreme sport with a high risk of serious injury;(10, 16-19)”
And a new section reading:
“Specifically, by nature of course design, DHI exposes cyclists to a high number and magnitude of accelerations; this can affect rider executive function in and of itself, but also creates an environment with high potential for SRC injuries.(9, 10)”
Methods:
The methods state a 6-step process was followed. However, only 5 stages are presented in the methods. Please could the authors clarify which is correct.
Apologies- there are 5 stages. The authors Arksey and O’Malley., (2005) outline an optional 6th step, which involves consulting stakeholders who can perhaps point out further references for inclusion or provide information on cost-effectiveness. It is known as a “consultation exercise”. However, we did not think this necessary for our review.
For the sake of clarity, we have changed the manuscript to read “5-step process”
Discussion:
L216 – It may be useful for the authors to clarify the typical duration of a DHI race, and hence the lack of utility for the SCAT-5 in such races.
Now reads:
“…which can render it non-viable during competition, particularly within a sport such as DHI (where races typically last 2-5 minutes total).(11)”
Please be consistent with the use of either RIDE-1 DMB or RIDE-DMB 1.
L264 – What/where was your last race? This may not be an appropriate question to ask, as the race official doing the assessment may not know what/where this was for that rider.
Now reads:
“What position are you seeded/what is your order in this race?”
L266 – Suggest removing the question ‘What is in your jersey (back) pockets?’ as DHI jerseys do not have pockets at all.
Now simply reads:
“What is your coach’s name?”
Based on current research by Hurst et al (2018; 2020) on the magnitude of head impacts in DHI and the potential to reduce cognitive function, could the authors provide some brief commentary on the potential role of technology in monitoring head impacts during DHI and its possible utility in decision making as to when to stop a rider for assessment?
The following section has been added, just before the conclusion.
“5.4. Future direction
With emergent evidence highlighting the magnitude of potential head impacts and their potential impairment of cognitive function (even without falling from the bike),(9, 10) future technology could incorporate accelerometers on helmets that sense when po-tential SRC events occur.
These could be applied in two ways. Firstly, pre-testing courses for DHI and limiting the number of rotational/translational accelerations that occur in any given race (or the total value in grams of significant accelerations (>10g)(10) to minimise acceptable injury risk may be an option; this would require further research into effects of potential SRC injury from cumulative low energy accelerations (with current evidence indicating re-duced executive functioning by this mechanism in DHI) and quantifying cut-offs.(9)
Alternatively, if cut-off/high-risk acceleration values were identified during races, this may provide evidence for removal of a cyclist where ambiguity exists, or encourage further assessment post-race.”
L275 – Should ‘HIA-DMB 1’ read ‘RIDE-DNB 1’?
Now reads RIDE-DHI.
Conclusions:
Please refer to RIDE-DMB to be consistent.
Now reads RIDE-DHI throughout.
Reviewer 2 Report
This paper provides an excellent scoping review of SRC in Downhill Mountain Biking, offering a needed addition to the literature on SRC in cycling. It is well written, clear and coherent. The practical recommendations and development of a framework for managing SRC in DMB is great to see and I’m sure this will have some good impact in the sport moving forward. I recommend this paper for publication and only have a few minor revisions to suggest to improve the paper. Firstly, I was surprised not to see the work of Hurst et al., (2018) and Hurst et al., (2020) mentioned in the review or introduction. I suggest incorporating this work into the paper as it offers an important contribution to the understanding of SRC in DMB, with a consideration for the head accelerations experienced by DMB riders in competition. Other minor suggestions have been included below.
- Hurst, H. T., Atkins, S., & Dickinson, B. D. (2018). The magnitude of translational and rotational head accelerations experienced by riders during downhill mountain biking. Journal of Science and Medicine in Sport, 21(12), 1256–1261. https://doi.org/10.1016/j.jsams.2018.03.007
- Hurst, H. T., Hancock, S., Hardwicke, J., & Anderson, E. (2020). Does participation in Downhill mountain biking affect measures of executive function?. Journal of Science and Cycling, 9(3), 74-83. https://doi.org/10.28985/1220.jsc.04
Abstract
Line 22: provide dates of the references
Line 28: The first line doesn’t read quite right. Perhaps ‘Concussion incidence was identified as between 5-23%.’ would work better.
Introduction
Line 55: it would help with clarity to state the incidence of concussion you are talking about. Is this in sport more broadly? Or in DMB? The reference used would suggest you are talking about sport more broadly.
Line 56: I’m not sure on this claim, why is the incidence likely to be increasing in cycling? This point needs better explanation and linking to the previous sentence.
Line 63: Change to (including the risk of Second Impact Syndrome)
Results
Line 185: Add date to Clark et al reference
Discussion
Line 209: I would suggest ‘message’ is more appropriate than ‘logo’ here. It may also be worth providing a brief description of this for readers unfamiliar with the ‘if in doubt, sit them out’ educational messaging, particularly readers outside of the UK.
Line 227: should this be ‘team or independent race doctors.’ ? Remove comma if so
Author Response
Reviewer 2
This paper provides an excellent scoping review of SRC in Downhill Mountain Biking, offering a needed addition to the literature on SRC in cycling. It is well written, clear and coherent. The practical recommendations and development of a framework for managing SRC in DMB is great to see and I’m sure this will have some good impact in the sport moving forward. I recommend this paper for publication and only have a few minor revisions to suggest to improve the paper. Firstly, I was surprised not to see the work of Hurst et al., (2018) and Hurst et al., (2020) mentioned in the review or introduction. I suggest incorporating this work into the paper as it offers an important contribution to the understanding of SRC in DMB, with a consideration for the head accelerations experienced by DMB riders in competition. Other minor suggestions have been included below.
- Hurst, H. T., Atkins, S., & Dickinson, B. D. (2018). The magnitude of translational and rotational head accelerations experienced by riders during downhill mountain biking. Journal of Science and Medicine in Sport, 21(12), 1256–1261. https://doi.org/10.1016/j.jsams.2018.03.007
- Hurst, H. T., Hancock, S., Hardwicke, J., & Anderson, E. (2020). Does participation in Downhill mountain biking affect measures of executive function?. Journal of Science and Cycling, 9(3), 74-83. https://doi.org/10.28985/1220.jsc.04
Thank you for referencing these papers- we believe they have improved the quality of our introduction.
Reference 9 is now: Hurst H, Hancock S, Hardwicke J, Anderson E. Does participation in Downhill mountain biking affect measures of executive function? Journal Of Science & Cycling. 2021;9:83-93.
Reference 10 is now: Hurst HT, Atkins S, Dickinson BD. The magnitude of translational and rotational head accelerations experienced by riders during downhill mountain biking. J Sci Med Sport. 2018;21(12):1256-61.
We have added the references at the following points:
“However, downhill racing is an arguably higher risk sport with regards to traumatic injury, and indeed head injury,(9, 10) but does not have any formal concussion policy.”
“Downhill mountain biking has previously been identified as an extreme sport with a high risk of serious injury;(10, 16-19)”
And a new section reading:
“Specifically, by nature of course design, DHI exposes cyclists to a high number and magnitude of accelerations; this can affect rider executive function in and of itself, but also creates an environment with high potential for SRC injuries.(9, 10)”
Abstract
Line 22: provide dates of the references
This has been corrected:
Arksey and O’Malley., (2005)
Levac et al., (2010)
Line 28: The first line doesn’t read quite right. Perhaps ‘Concussion incidence was identified as between 5-23%.’ would work better.
Thank you. We have made this amendment.
Introduction
Line 55: it would help with clarity to state the incidence of concussion you are talking about. Is this in sport more broadly? Or in DMB? The reference used would suggest you are talking about sport more broadly.
Line 56: I’m not sure on this claim, why is the incidence likely to be increasing in cycling? This point needs better explanation and linking to the previous sentence.
Line 55+56 now read:
The exact incidence of concussion in sport generally is poorly defined as many cases are thought to go unreported.(13) It is likely, at least partially for this reason, that the incidence is increasing in cycling alongside an increased awareness and assessment for concussion, which has one of the highest incidence of SRC of any sport.(14, 15)
Thank you for identifying this.
Line 63: Change to (including the risk of Second Impact Syndrome)
We have changed this to incorporate your suggestion.
Results
Line 185: Add date to Clark et al reference
This has been amended.
Discussion
Line 209: I would suggest ‘message’ is more appropriate than ‘logo’ here. It may also be worth providing a brief description of this for readers unfamiliar with the ‘if in doubt, sit them out’ educational messaging, particularly readers outside of the UK.
Now reads:
“…and the message, ‘if in doubt, sit them out’.(35) This message encourages sports teams, athletes and coaches to err on the side of caution, removing an athlete from play in any event where an SRC has even potentially occurred.”
Line 227: should this be ‘team or independent race doctors.’ ? Remove comma if so
Now reads:
“…assessed on the side-line by key team staff or independent race doctors”